# Zero-shot Cross Language Text Classification

## Abstract

Labeled text classification datasets are typically only available in a few select languages. In order to train a model for e.g news categorization in a language $L_t$ without a suitable text classification dataset there are two options. The first option is to create a new labeled dataset by hand, and the second option is to transfer label information from an existing labeled dataset in a source language $L_s$ to the target language $L_t$. In this paper we propose a method for sharing label information across languages by means of a language independent text encoder. The encoder will give almost identical representations to multilingual versions of the same text. This means that labeled data in one language can be used to train a classifier that works for the rest of the languages. The encoder is trained independently of any concrete classification task and can therefore subsequently be used for any classification task. We show that it is possible to obtain good performance even in the case where only a comparable corpus of texts is available.

## 1 Introduction

Automatic systems that can classify documents quickly and precisely are useful for a wide range of practical applications. For example, organizations may be interested in using *sentiment analysis* of opinion posts such as tweets that mention their products and services. By classifying the sentiment of each post (e.g. positive, neutral, or negative), the organization can for example learn which parts of a product should be improved.

Creating a suitable, large labeled dataset for training a classification model requires a lot of effort and available public datasets are typically only available in the most common languages. In order to train a classification model for a languages $L_t$ without a suitable text classification dataset there are two options: The first option is of course to create a new labeled dataset from scratch, and the second option is to use the label information in existing labeled datasets in a language $L_s$ and then transfer this label information to $L_t$. The first option usually requires a great amount of work and is typically not a viable solution. The second option is called cross-language text classification (CLTC) (Wan, 2009).

In this article we present a method for performing CLTC by means of a *universal encoder*. The method consists of two steps. In the first step, a *universal encoder* is trained to give similar representations to texts that describe the same topic, even if the texts are in different languages. In the second step, a classification module uses the language-independent representations from the universal encoder as inputs and is trained to predict which category each document belongs to. Compared to previous work, this method has several advantages:

1. The universal encoder can be trained using just a *comparable corpus*. A comparable corpus is a corpus where the multilingual versions of a document are not necessarily translations of each other, but merely about the same topic.

2. It enables zero-shot classification. I.e. if we have a comparable corpus in French-Spanish, we can build a classifier for Spanish by using a labeled dataset in English.

3. The universal encoder does not rely on single word translations, but rather on encoding entire contexts. This can help alleviate ambiguity problems caused by polysemy.

4. The input language does not have to be specified at test time.

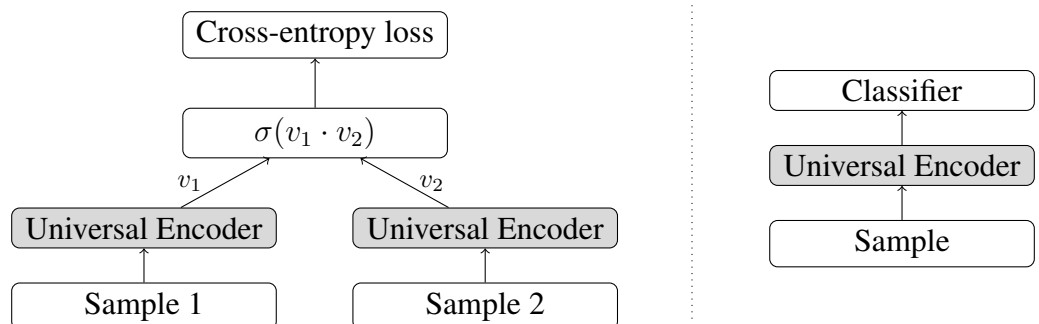

Figure 1: Illustration of how the *Universal Encoder* is trained (left) and how it is used when training and making predictions using the classifier (right). All *Universal Encoders* (gray background) are exactly identical by sharing architecture and weights

The method presented in this article is conceptually similar in spirit to Google's zero-shot machine *translation* model (Johnson et al., 2016), which is used in the Google translate API. That model also uses a shared vocabulary and a language independent encoder. It does, however, require a large corpus of aligned sentences for training. Additionally, translating a text is a much harder problem than merely extracting discriminative features since it requires encoding of e.g. syntactic information that is not necessary for text classification. Therefore such a model is much more complex than it needs to be, and a more parsimonious model is therefore preferable. We will compare our zero-shot *classification* model with an equivalent model based on the zero-shot *translation* model in section 3.

The rest of the article is organized as follows: We present our CLTC model in section 2. Experiments, data and results are presented in section 3. In section 4 we review previous approaches to cross-lingual text classification. In section 5 we will take a look at some possible improvements and future directions for the method. Finally, we conclude the article in section 6.

## 2 ALIGNMENT OF DOCUMENT REPRESENTATIONS

The zero-shot classification model consists of two independent components, a universal encoder and a classifier module. The two components will be described in the following.

### 2.1 UNIVERSAL ENCODER

The universal encoder provides a language independent encoding of all information useful in order to identify a text. If the language independent representation is very close for comparable texts, a classifier for representations in one language should be usable across all languages.

The universal encoder transforms the input text by using a function $F_u$. The mapping $F_u$ could for example consist of a word embedding followed by a recurrent layer, or a sum of word embedding vectors. Or it could be the LSA representation of the input text followed by a dense layer.

To create a training sample for the encoder, a pair of texts are drawn from the corpus. Half of the pairs will consist of two texts on the same topic (but in different languages) and half the pairs will consist of two texts on different topics (and possibly different languages). The encoder is trained (using cross-entropy error) to be able to predict whether two texts $s_1$ and $s_2$ are comparable (i.e. about the same topic) based on the inner product $F_u(s_1) \cdot F_u(s_2)$ between their representations. The encoder training setup is illustrated on the left side of fig. 1.

### 2.2 CLASSIFIER

The output of the universal encoder is a language independent representation of a text. Training a classifier based on the universal encoder will give a universal classifier that can be used to classify text in any language that the encoder has been trained on. The universal classifier is depicted on the right hand side of fig. 1.

## 3 EXPERIMENTS

We compare our zero-shot classification model with two other models on the task of predicting the category of Italian Wikipedia articles.

- **A monolingual classifier**. The performance of this model can be considered an upper bound for the performance. However, in a realistic scenario we would only have a very limited number of native samples available. We therefore train the model several times using a varying number of native samples from as little as 5k to more than a hundred thousand. This will give the performance of the native classifier as a function of number of available native samples, and will enable us to tell how many native samples are required in order to obtain a performance equivalent to the that of the CLTC models.

- **A model based on machine trainslation.** As mentioned, Google's zero-shot translation model is conceptually similar to our model. It is therefore natural to compare a model based on that model with the zero-shot classification model presented in this paper.

### 3.1 DATA

We evaluate the universal encoder using Wikipedia article abstracts in Italian, German, French and English. Wikipedia inter-language links are used to relate articles in different languages about the same topic (such as "Tom Cruise"). We use DBPedia mapping-based properties[1] to assign each topic to a category (such as Person or City). For pages with multiple categories we select one at random. We restrict the number of classes to the 200 most frequent ones. The number of articles in each language can be seen in table 1. We don't do any pre-processing besides removing punctuation characters (such as `., : | −_/ " * ' =`) and tokenizing the text.

Table 1: Number of articles in different languages in the dataset. There are on average 2.3 articles for each Wikipedia topic.

| Italian | 493k |
|---------|------|
| German | 675k |
| French | 796k |
| English | 1457k |

The dataset is split evenly into an *encoder dataset* and a *classification dataset*. The encoder dataset is used to train the universal encoder and the classification dataset is used to train the different classification models. When training the encoder, we sample uniformly between pairs of articles with the same topic and pairs of articles with different topic. In the first column of table 2 we see the number of training pairs for the encoder for the different language sets. Column two shows the number of samples in the classifier training set for the different language pairs (i.e. the number of non-Italian articles in the classification dataset). Column three shows the number of samples in the classifier test set for the different language sets (i.e. the number of Italian articles in the classifier dataset). Topics with only one article are removed, and so are topics not belonging to one of the 200 target classes.

It is important to note that the classification training dataset for our classification model *does not* include any Italian samples.

Table 2: Number of samples for the different language sets. The number of training samples for the encoder denotes the number of pairs of articles about the same topic but in different languages.

| | Encoder train pairs | Classifier train samples | Classifier test samples |
|----------|--------------------|------------------------|------------------------|
| It-En | 245k | 163k | 167k |
| It-En-Fr | 813k | 864k | 167k |
| It-En-Fr-De | 1500k | 1305k | 167k |

The Wikipedia articles are by no means translations of each other. Typically the English articles are much longer compared to articles in other languages and the content is often quite different. We

---

[1]See http://wiki.dbpedia.org/services-resources/datasets/dbpedia-datasets#h434-10 for a description.

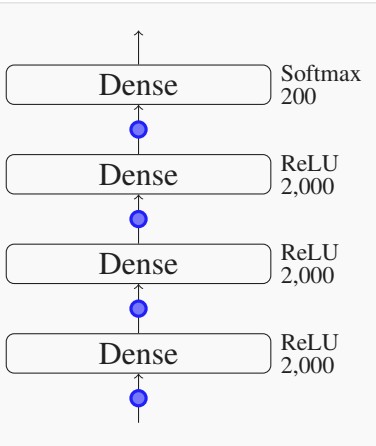

**Classification module**

**Encoder module**

(a) The architecture of the encoder.

(b) The architecture of the classification module of the neural models. The numbers to the right of each layer is the activation function of the layer (above) and the number of units (below). The blue dots represent dropout layers with 50% dropout probability.

therefore use the heuristic that the *essence* of an article is contained in its beginning. As a consequence, the semantic content of the beginnings of the articles in different languages is hopefully more similar than the entire texts. For this reason we use only the first 200 words of each article.

We do not use the entire remaining article text as input samples. Instead we draw a random *snippet* from the text. A snippet from a text document is a small sequence of random length consisting of 3-200 tokens. Note that the snippet idea can be seen as an extreme form of input dropout (we dropout everything but a small fraction of the text). That is, the snippet sampling provides a degree of regularization.

### 3.2 MODELS

The architectures of the three models have been kept as similar as possible in order to provide the most fair comparison. We use Adam updates (Kingma & Ba, 2014) with default parameters (except for a learning rate set to $10^{-4}$) for training of all of the models. The architectures of the different models will be described in the following.

#### 3.2.1 ZERO-SHOT CLASSIFICATION MODEL

The architecture of the universal encoder can be seen in fig. 2a. The embedding layer is shared among all input languages and therefore the vocabulary can become quite large. As we will see in section 3.3, it is beneficial to choose the embedding vector size as large as possible. We therefore use a word based hash embedding in the input layer (Svenstrup et al., 2017). A hash embedding is a recently proposed improvement to a regular embedding that requires far less parameters for large vocabulary problems or problems with large embedding sizes. It is thus perfect for our purpose. We use a hash embedding with 25k buckets and two hash functions for all experiments. The embedding vector size is varied between 250 and 1500 in the experiments.

Following the embedding layer we have a sum layer where the coordinate-wise sum of all word vectors is computed. On top of the sum layer we use a dense layer with twice the number of units as the embedding vector size.

Once the encoder has been trained we freeze the parameters and use the universal encoder as input to the classification module (see right side of fig. 1). The architecture of the classification module is illustrated in fig. 2b. It is a neural network consisting of three dense layers with 2000 units with rectified linear units activation, followed by a dense layer with 200 units (number of classes) with

softmax activation. We use dropout (Srivastava et al., 2014) with a $50\%$ dropout probability as regularization between all the layers.

### 3.2.2 NATIVE CLASSIFIER

The architecture of the native classifier is identical to the architecture of the universal encoder + classifier described above. The differences are that we train the encoder and classifier simultaneously, and that the model is both trained and tested on Italian samples. It is trained on 3.5k, 17k, 34k, 68k, 137k and 158k samples. It is tested on 8k Italian samples.

### 3.2.3 CLASSIFIER BASED ON MACHINE TRANSLATION

The architecture of this model is identical to the architecture of the native classifier. The differences are that we first train on all *English* samples in the classification dataset and then test on a testset of Italian samples translated to English by using Google translate.

### 3.3 RESULTS

### 3.3.1 THE UNIVERSAL ENCODER

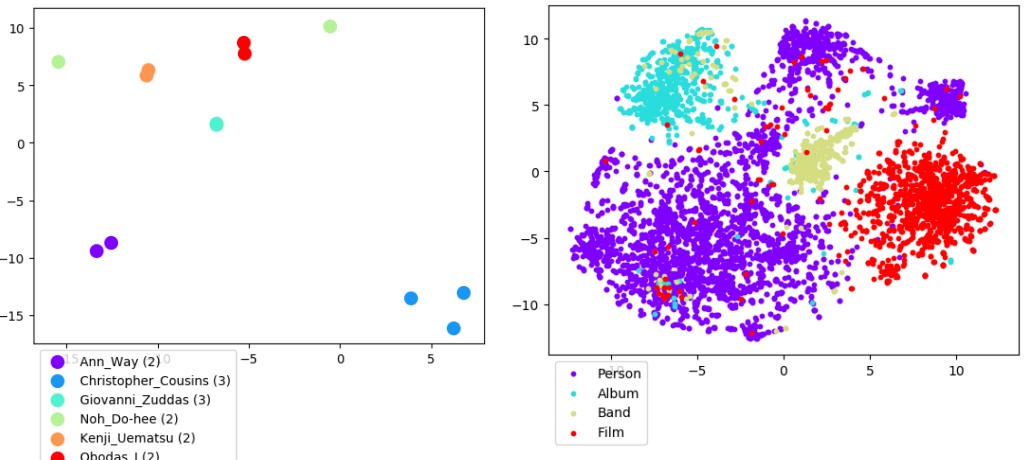

(a) t-SNE plot of universal representations of articles about persons. The articles are from the classifier dataset so the encoder has not been trained on the articles.

(b) t-SNE plot of universal representations of multilingual articles of different categories. The articles are from the classifier dataset and the encoder has therefore never seen the the documents before. The plot shows good separation between the categories.

The purpose of the universal encoder is to provide representations for articles such that articles about *different* topics are distant from each other, while articles on the *same* topic are almost identical, even if the articles are in different languages. Figure 3a displays a t-SNE plot (Maaten & Hinton, 2008) of the universal representations of a few articles about persons. The plot shows good separation between articles on different people, and a low separation between articles about the same person in different languages.

The t-SNE plot in fig. 3b shows universal representations of articles from multiple different languages and categories. Even though the encoder wasn't ever explicitly trained for that purpose, it easy to see the articles from different categories are nicely separated even though some of the categories are very similar.

### 3.4 CLASSIFICATION RESULTS

We report the accuracy for the different models in table 3. We see that the zero-shot classifier obtains very good results, especially in a bilingual setting where the largest classifier (embedding size 1500)

obtains a very high accuracy of 78.5%. This is far better than the model based on machine translation (72.1%) and very close to the upper bound on accuracy of 80.4% (the native classifier).

In fig. 4a we see accuracy as a function of the embedding size for the different language sets. We note that the accuracy increases monotonically with the embedding size for all language sets. However, this increase seems to be less pronounced for larger embedding sizes. We also note that the accuracy unfortunately decreases with the number of languages. The reason for this decrease in performance is less clear and does not seem to be caused by lack of capacity in the encoder. We believe that it may be caused by the sampling method used. We sample uniformly from topics and then select a random language pair. Since there are much more articles in English, French and German than in Italian (see table 1), there will be many more sampled pairs from these languages. This may cause the encoder to favour these languages. We have, however, not investigated this further.

Table 3: Test accuracy [%] of the different classification models. ZSC denotes our zero-shot classification model. All results are with an embedding size of 1500. Since the dataset is not balanced we have included a largest class classifier in the results.

| Model | Accuracy |
|---|---|
| ZSC (It-En) | 78.5 |
| ZSC (It-En-Fr) | 75.9 |
| ZSC (It-En-Fr-De) | 73.5 |
| Machine translation | 72.1 |
| Native classifier | 80.4 |
| Largest class | 9.0 |

A typical (realistic) scenario for use of CLTC methods is where a small amount of native, labeled data is available, but where a large comparable corpus is available. In that case our zero-shot classification model will have a higher performance compared to the monolingual native classifier. As we see in fig. 4b the native monolingual classifier needs about 100k native samples in order to obtain a higher performance compared to our zero-shot classification model.

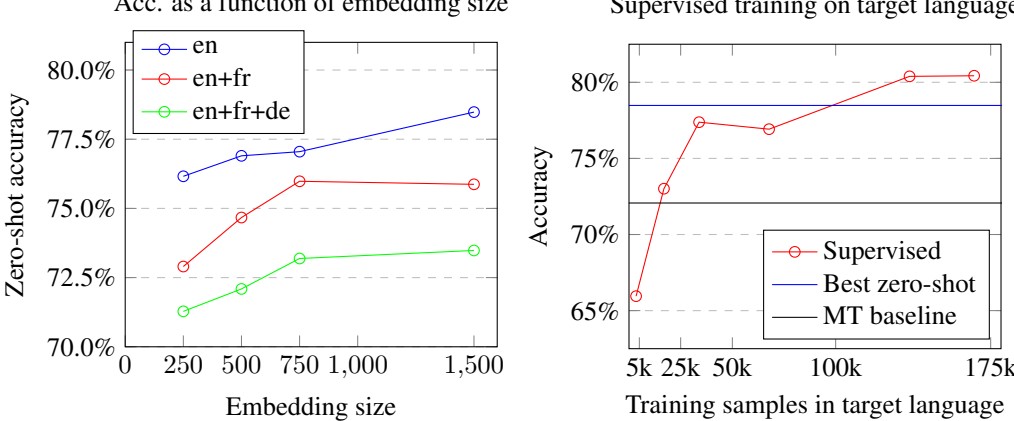

(a) Accuracy as a function of embedding size for the different language sets.

(b) Accuracy as a function of training samples in target language (Italian) for a native classifier

## 4 RELATED WORK

Several strategies for multilingual text classifications have previously been proposed. The different strategies can be grouped into three groups according to whether they require a corpus of aligned sentences (a parallel corpus), a comparable corpus or just a dictionary. The different approaches will be described in the following.

### 4.1 Approaches requiring a parallel corpus

The main problem with these methods is of course that an aligned corpus is typically only available for a handful of language pairs. One such corpus is the Europarl dataset for machine translation (Koehn, 2005), which is available in 21 European languages.

#### 4.1.1 Multilingual word embeddings

A word embedding is a mapping from a set of words to a set of dense real-valued vectors. Such representations of words have proven to be very useful in many monolingual natural language processing problems. By exploiting that words occurring in the same type of context have similar meaning, it is possible to perform unsupervised training of word embeddings such that words with the same meaning have similar vector representation (Bengio et al., 2003). In (Klementiev et al., 2012; Chandar et al., 2014), this property is extended to a multilingual setting such that words with the same meaning, but possibly in different languages, have similar representation. Klementiev et al. (2012) uses a multitask learning algorithm for training the embeddings, and Chandar et al. (2014) uses a method based on autoencoding.

Once created, the multilingual word embeddings can be used to perform cross-lingual text classification. Note that just as the universal encoder introduced in this article, the word embeddings are not trained with any specific classification task in mind. I.e. the word embeddings can be trained once and subsequently be used in a variety of classification tasks.

Multilingual word embeddings can also be trained using only a dictionary, see below.

#### 4.1.2 Approaches based on machine translation

A lot of work in cross-language text classification has relied on the availability of machine translation models. Most of the methods use a two step process where features are either extracted and then translated (Shi et al., 2010; Montalvo et al., 2007; Wei & Pal, 2010), or are extracted from the translated text (Wan, 2009; Ling et al., 2008; Rigutini et al., 2005).

Methods based on machine translation are attractive because they are typically intuitive and easy to understand. Unfortunately, the machine translation step introduces a lot of noise in the form of information loss, translation error and noise due to the discrepancy between data distributions of the different languages. Several methods have been proposed in order to reduce the performance penalty induced by the translation step. These methods include model translation based on the EM algorithm (Rigutini et al., 2005; Shi et al., 2010), and methods based on domain adaption (Wei & Pal, 2010; Blitzer et al., 2006)

### 4.2 Approaches requiring a comparable corpus

A comparable corpus is typically much easier to obtain than a parallel corpus since it merely requires that the topic of a text is known across languages. For example, news articles are typically tagged with categories such as finance or sports and these categories can be used to create a comparable corpus for news classification. Wikipedia is another very good example of a comparable corpus.

#### 4.2.1 LDA approaches

Latent Dirichlet Allocation (LDA) is a Bayesian Network model where each document is assumed to have a latent topic distribution. For each topic there is a corresponding word distribution. In order to create a document of $N$ words, we first draw a topic distribution $T_d$ for the document. Each of the $N$ words in the document is then generated by first drawing a word topic from $T_d$, and then use the word distribution corresponding to the topic to generate the word. This can easily be extended to a multilingual setting by letting documents with the same content (but in different languages) share the same topic distribution. The topic distribution of a document can then be estimated after training. Documents with similar topic distribution will tend to be semantically similar. This property can be used to train cross-lingual classifiers. Multilingual LDA approaches to cross-language text classification have been explored in (De Smet et al., 2011; Ni et al., 2011).

### 4.2.2 MULTI-VIEW LEARNING

In multi-view learning it is assumed that different language versions of a document describe the same data object. The representation of the views should therefore be similar. There are several variants of the multi-view learning method but typically the method consists in optimizing a set of monolingual classifiers subject to the constraint that the representation of the views should be similar (Amini & Goutte, 2010; Wan, 2009; Guo & Xiao, 2012). The different views are often constructed using machine translation, however.

## 4.3 APPROACHES REQUIRING ONLY A DICTIONARY

These approaches are attractive due to the low requirement on data alignment. However, such methods have an inherent problem with poly-synonymous words since they rely on single word translations that ignore the context.

### 4.3.1 STRUCTURAL CORRESPONDENCE LEARNING

In structural correspondence learning (SCL) (Blitzer et al., 2006; Prettenhofer & Stein, 2010) a set of discriminative words called pivots are identified in a source language and translated to a target language. Each pivot (and its translation) will induce a bisection of the union of the texts in target and source languages. For each pivot a simple linear classifier is trained to predict if a text (with all occurrences of the pivot deleted) contains the pivot. The information contained in the parameters of all of the classifiers are then used to create a bilingual classifier. SCL can be trained using just a corpus of labeled data in a source language, translations of the pivots and an unlabeled corpus for the target language. SCL has shown a performance equal to that of models based on machine translation (Blitzer et al., 2006).

### 4.3.2 MULTILINGUAL WORD EMBEDDINGS

Multilingual word embeddings can also be trained using only a dictionary and unlabeled text (Wick et al., 2016). This can be done by switching words with the same meaning in different languages. E.g. the sentence *The red hand* could be modified to *The rojo hand* using a dictionary. These artificially modified sentences can then be used to train multilingual word embeddings by using e.g. a CBOW (Mikolov et al., 2013) model. There are some challenges to using this kind of model, however. For example, the method relies on the fact that the meaning of a word is determined by its context (the distributional hypothesis). But in the example above readers familiar with Spanish grammar rules will know that rojo and hand would not belong to the same context (but roja and hand would).

## 5 FUTURE WORK

There are several interesting directions for future work. First of all, it would be interesting to experiment with more complex versions of $F_u$ such as recurrent nets.

In the experiments presented in the article we only did a small amount of experimentation with e.g. the size of the embedding layer and the transformation to a language independent representation. We believe that even better results may be obtained by hyper-parameter optimization.

In this article we used Wikipedia both when training the encoder and the classifier. However, if the discriminative features in the encoder corpus is sufficiently close to the discriminative features in the classifier corpus it is perhaps possible to use an encoder corpus that is different from the classifier corpus. E.g. we could train the encoder on Wikipedia articles and then use the encoding to classify news articles.

Our results show that performance decreases with the number of languages. As mentioned in section 3.3, this might be caused by the sampling method. It would be interesting to test this hypothesis by super sampling the smaller languages.

Finally, hinge loss would probably have been a more natural loss function for the encoder instead of cross-entropy loss. It could be interesting to see if performance could be improved by changing the loss function.

## 6  CONCLUSION

In this article we have shown how to create a language independent representation using only a corpus of comparable texts. The language independent representation can subsequently be used for zero-shot classification.

We show that it is possible to obtain very good performance even when only a comparable corpus of texts is available.

The unsupervised classifier of course does not perform better than a supervised classifier trained on the same number of samples. It is, however, equal in performance to a native language supervised classifier trained on about hundred thousand samples. This means that if the number of native samples is limited and a large comparable corpus is available, the performance of our zero-shot classification can be better than that of a monolingual classifier.

Our results show that even though it is possible to obtain good results using several languages at once, the best performance is obtained by using only two languages. Our results furthermore show that it is necessary to use a very large embedding size in order to obtain the best possible performance.

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
