# OpenReview forum: "Zero-shot Cross Language Text Classification"
_ICLR.cc/2018/Conference — Reject_

### Official Review · AnonReviewer2 · 2017-11-27
**Method is not sufficiently presented and empirical results are less convincing.**

**Rating:** 4
**Confidence:** 3

**Review:**

This paper proposes a language independent text encoding method for cross-language classification. The proposed approach demonstrates better performance than machine translation based classifier.

The proposed approach  performs language independent common representation learning for cross-lingual text classification. Such representation learning based methods have been studied in the literature. The authors should provide a review and comparison to related methods.

Technical contribution of the paper is very limited. The approach section is too short to provide a clear presentation of the model. Some descriptions about the input text representation are actually given in the experimental section.

The proposed approach uses comparable texts across different languages to train the encoders, while using the topic information as auxiliary supervision label information. In the experiments, it shows the topics are actually fine-grained class information that are closely related to the target class categories. This makes the zero-shot learning scenario not to be very practical. With such fine-grained supervision knowledge, it is also unfair to compare to other cross-lingual methods that use much less auxiliary information.

In the experiments, it states the data are collected by “For pages with multiple categories we select one at random”.  Won’t this produce false negative labels on the constructed data? How much will this affect the test performance?

The experimental results are not very convincing without empirical comparisons to the state-of-the-art cross-lingual text classification methods.

---

### Official Review · AnonReviewer3 · 2017-11-27
**Simplistic model, faulty writeup, thin experiment**

**Rating:** 2
**Confidence:** 4

**Review:**

The draft proposes an approach to cross-lingual text classification through the use of comparable corpora, as exemplified through the use of Wikipedia via the inter-language links. A single task is featured: the prediction of categories for the Italian Wikipedia articles. Two models are contrasted to the proposed zero-shot classification approach, a monolingual classifier and a machine translation-based model.

I have a number of issues with the paper, and for these I vote strong reject. I briefly list some of these issues.

1) The model brings no novelty, or to put it bluntly, it is rather simplistic. Yet, at the same time, its description is split over multiple sections and thus rather convoluted, in effect obscuring the before-mentioned over-simplicity.
2) The experiment is also oversimplified, as it features only one target language and a comparison to an upper bound and just a single competing system.
3) In contrast to the thin experiments and (lack of) technical novelty, the introduction & related work writeups are overdrawn and uninteresting.

I am sorry to say that I have learned very little from this paper, and that in my view it does not make for a very compelling ICLR read.

---

### Official Review · AnonReviewer1 · 2017-11-29
**Simple idea; experimental design could use improvements**

**Rating:** 3
**Confidence:** 4

**Review:**

This paper addresses the problem of learning a cross-language text categorizer with no labelled information in the target language. The suggested solution relies on learning cross-lingual embeddings, and training a classifier using labelled data in the source language only.

The idea of using cross-lingual or multilingual representations to seamlessly handle documents across languages is not terribly novel as it has been use in multilignual categorization or semantic similarity for some time. This contribution however proposes a clean separation of the multiligual encoder and classifier, as well as a good (but long) section on related prior art.

One concern is that the modelling section stays fairly high level and is hardly sufficient, for example to re-implement the models. Many design decisions (e.g. #layers, #units) are not justified. They likely result from preliminary experiments, in that case it should be said.

The main concern is that the experiments could be greatly improved. Given the extensive related work section, it is odd that no alternate model is compared to. The details on the experiments are also scarce. For example, are all accuracy results computed on the same 8k test set? If so this should be clearly stated. Why are models tested on small subsets of the available data? You have 493k Italian documents, yet the largest model uses 158k... It is unclear where many such decisions come from -- e.g. Fig 4b misses results for 1000 and 1250 dimensions and Fig 4b has nothing between 68k and 137k, precisely where a crossover happens.

In short, it feels like the paper would greatly improve from a clearer modeling description and more careful experimental design.

Misc:
- Clarify early on what "samples" are in your categorization context.
- Given the data set, why use a single-label multiclass setup, rather than multilabel?
- Table 1 caption claims an average of 2.3 articles per topic, yet for 200 topics you have 500k to 1.5M articles?
- Clarify the use of the first 200 words in each article vs. snippets
- Put overall caption in Figs 2-4 on top of (a), (b), otherwise references like Fig 4b are unclear.

---

### Decision · Program_Chairs · 2018-01-29
**ICLR 2018 Conference Acceptance Decision**

**Decision:**

Reject

**Comment:**

Unfortunately, it falls short of ICLR standards -- from evaluation, novelty and clarity perspectives. The method is also not discussed in all details.